# *het-B* allorecognition in *Podospora anserina* is determined by pseudo-allelic interaction of genes encoding a HET and lectin fold domain protein and a PII-like protein

**Corinne Clavé**[1◉], **Sonia Dheur**[1◉], **Sandra Lorena Ament-Velásquez**[2],
**Alexandra Granger-Farbos**[1], **Sven J. Saupe**[1]*

**1** IBGC, UMR 5095, CNRS-Université de Bordeaux, Bordeaux, France, **2** Division of Population Genetics, Department of Zoology, Stockholm University, Stockholm, Sweden

◉ These authors contributed equally to this work.
* sven.saupe@ibgc.cnrs.fr

**Data Availability Statement:** Sequence data is deposited in Genbank (accession number: OR910067).

## Abstract

Filamentous fungi display allorecognition genes that trigger regulated cell death (RCD) when strains of unlike genotype fuse. *Podospora anserina* is one of several model species for the study of this allorecognition process termed heterokaryon or vegetative incompatibility. Incompatibility restricts transmission of mycoviruses between isolates. In *P. anserina*, genetic analyses have identified nine incompatibility loci, termed *het* loci. Here we set out to clone the genes controlling *het-B* incompatibility. *het-B* displays two incompatible alleles, *het-B1* and *het-B2*. We find that the *het-B* locus encompasses two adjacent genes, *Bh* and *Bp* that exist as highly divergent allelic variants (*Bh1/Bh2* and *Bp1/Bp2*) in the incompatible haplotypes. *Bh* encodes a protein with an N-terminal HET domain, a cell death inducing domain bearing homology to Toll/interleukin-1 receptor (TIR) domains and a C-terminal domain with a predicted lectin fold. The *Bp* product is homologous to PII-like proteins, a family of small trimeric proteins acting as sensors of adenine nucleotides in bacteria. We show that although the *het-B* system appears genetically allelic, incompatibility is in fact determined by the non-allelic *Bh1/Bp2* interaction while the reciprocal *Bh2/Bp1* interaction plays no role in incompatibility. The highly divergent C-terminal lectin fold domain of BH determines recognition specificity. Population studies and genome analyses indicate that *het-B* is under balancing selection with trans-species polymorphism, highlighting the evolutionary significance of the two incompatible haplotypes. In addition to emphasizing anew the central role of TIR-like HET domains in fungal RCD, this study identifies novel players in fungal allorecognition and completes the characterization of the entire *het* gene set in that species.

## Author summary

Many cellular life forms display genetic systems that protect individuality and discriminate conspecific self from non-self. In filamentous fungi, cell fusion events between strains

**Funding:** S.L.A.-V would like to acknowledge support from the Swedish Research Council (grant 2022-00341) The SJS lab is supported by funding from the CNRS and University of Bordeaux. The funders had no role in study design, data collection and analysis, decision to publish, or preparation of the manuscript.

**Competing interests:** The authors have declared that no competing interests exist.

are under check by specific allorecognition genes that trigger regulated cell death upon detection of non-self. The role of incompatibility is to restrict mycovirus transmission and conspecific parasitism. *Podospora anserina*, a good model for the study of this form of allorecognition, harbors nine incompatibility *het* loci. Previous studies have revealed that these genes can be homologous to genes with immune functions in other phyla including bacteria, plants and animals. We have cloned *het-B*, the last of the nine *het* genes that remained to be identified and found that it is a complex locus comprising two adjacent genes *Bh* and *Bp*. BH displays an N-terminal HET domain (related to TIR domains) and a C-terminal domain with a predicted lectin fold. BP is homologous to PII-like proteins, known bacterial metabolite sensors. Intriguingly, despite apparent genetic allelism, incompatibility is dictated by the non-allelic *Bh/Bp* interaction. This study stresses the reoccurring involvement of HET domains in fungal RCD and signs completion of the characterization of the entire set of *het* loci in that species, enabling a comparative analysis of the different genetic architectures underlying allorecognition.

## Introduction

Allorecognition, the ability to discriminate conspecific self from non-self has been described in a variety of cellular life forms including myxobacteria, slime molds, colonial cnidaria, uro-chordates and mammals [1–5]. In fungi, a specific form of allorecognition known as hetero-karyon incompatibility occurs during somatic cell fusion events. Incompatibility corresponds to the onset of regulated cell death (RCD) when cells from genetically distinct individuals fuse [6–9]. This form of allorecognition is determined by so-called *het* genes (also termed *vic* in some species). Spontaneous ability for somatic cell fusion and syncytial structure make the fungal mycelium vulnerable to viral infections and conspecific parasitism. Incompatibility was found to limit the transmission of mycoviruses and other deleterious replicons between strains [6,10]. Incompatibility could also protect fungal individuality by avoiding exploitation of the mycelial resources by non-adapted parasitic nuclei [11]. *Het* genes in fungal populations are generally under frequency-dependent balancing selection and show trans-species polymor-phism, that is, the retention of polymorphic variants across several speciation events [12–17]. Trans-species polymorphism has been described for allorecognition loci in other phyla such as the vertebrate MHC [18,19], self-incompatibility loci in flowering plants [20], or social allore-cognition loci in ants [21].

In the Sordariomycete *Podospora anserina*, genetic analysis of a collection of 15 isolates originating from various sites in France led to the identification of a total of nine incompatibil-ity loci (S1 Fig) [22]. Genetically, incompatibility systems have been categorized into allelic sys-tems in which incompatibility is triggered by gene interactions of alleles of the same locus (*het-B*, *het-V*, *het-Q*, *het-S*, and *het-Z*) and non-allelic systems involving interaction of specific alleles of two distinct loci (*het-C/het-E*, *het-C/het-D* and *het-R/het-V*). *Het-V* is hence involved both in allelic and non-allelic interactions [9,13] (S1 Fig). *Het* loci are generally biallelic, although a more complex situation with multiallelic loci is encountered for *het-C*, *D*, and *E* [23]. A genetic difference at any of those loci is sufficient to trigger incompatibility. That is, two strains must have compatible allele combinations at all nine loci to form viable heterokary-ons, so that in practice confrontation between natural isolates most often result in incompati-bility [13,22]. In *P. anserina*, as in many other species, incompatibility can be easily discerned macroscopically when strains are confronted on solid media. Regulated cell death in the con-frontation zone leads to the formation of an abnormal, dense and opaque contact line, termed

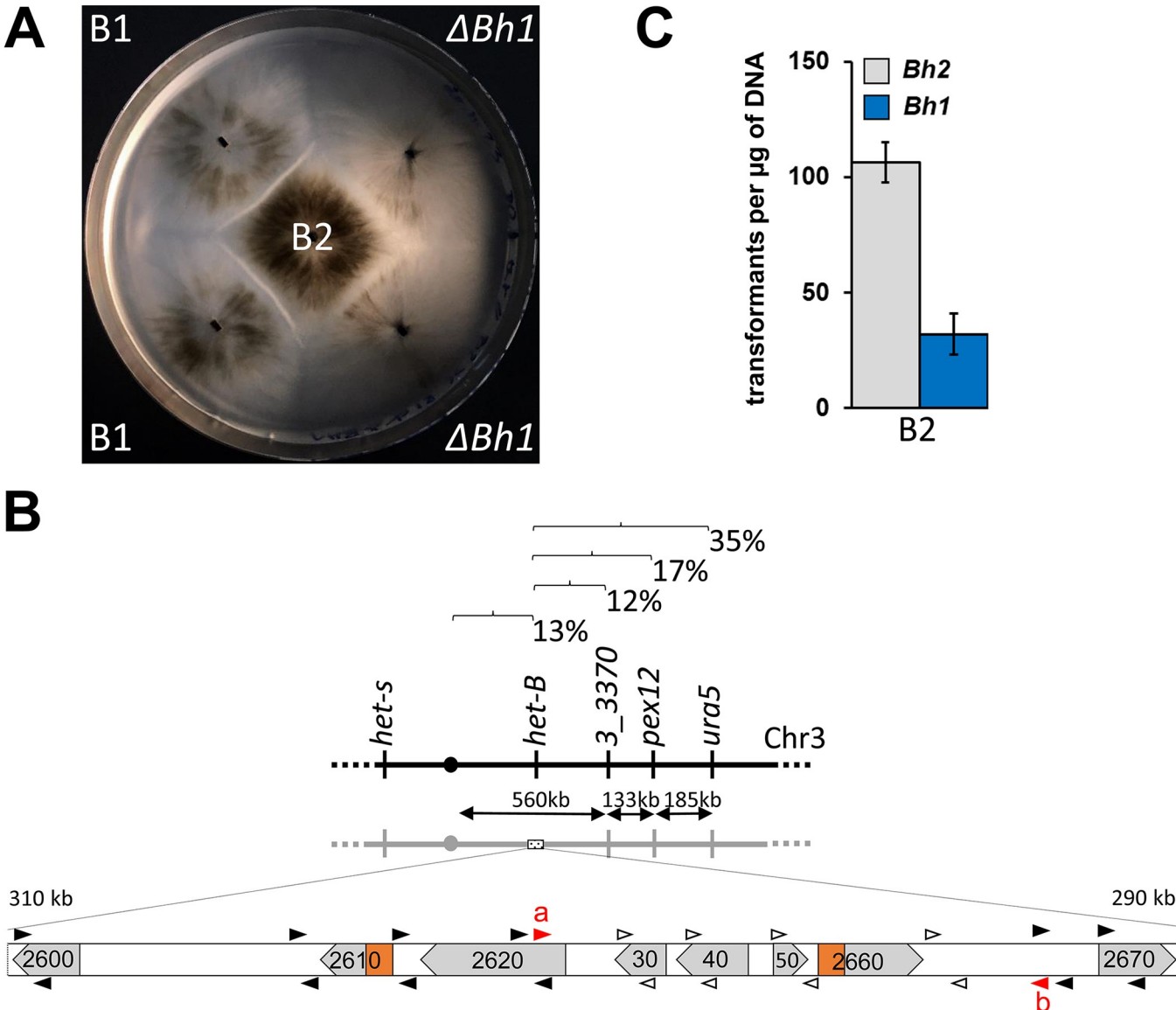

**Fig 1. Genetic mapping and characterization of the *het-B* locus.** (A) Barrage formation (incompatibility) between B1 and B2 and loss of incompatibility in *ΔBH1/B2* confrontations. (B) Genetic map of the *het-B* region. Recombination frequencies between the markers are given together with physical distances. Below an ORF map of a 20 kb region spanning the expected position of the *het-B* locus is given. The gene designations have been simplified by omitting the Pa_3_ or Pa_3_26 prefix. Regions encoding HET domains are marked in orange. Results of the PCR-mapping of the B1/B2 polymorphism are given with primer pairs leading to amplification in both backgrounds represented as black arrowheads and primer pairs amplifying only in the B1 background as open arrowheads. Primers a and b given in red were used to amplify the entire haplotype in B1 and B2 strains. (C) Transformation efficiency of plasmids bearing *Bh1* or *Bh2* into B2 protoplasts. Results are triplicates with standard deviation.

barrage (Fig 1A). During the sexual cycle, heteroallelism at some *het* loci also leads to pleiotropic pre- and post-zygotic effects ranging from partial sterility, formation of self-incompatible progeny, and meiotic drive in the form of spore-killing [13,24,25] (S1 Fig). Consequently, *het* gene polymorphism in populations has significant evolutionary consequences on outcrossing, gene flow and speciation [13, 26]. Cloning of the *P. anserina het* loci has been progressively undertaken using functional or positional cloning and candidate gene approaches. These studies led successively to the molecular characterization of *het-S* [27], *het-C* [28], *het-E* [29], *het-D* [30], *het-R* [31], *het-Z* [14], *het-Q* [32] and *het-V* [13]. *Het* genes have also been isolated in

*Neurospora crassa* [14,17,33–38] and *Cryphonectria parasitica* [39,40], and more recently in *Aspergillus fumigatus* [16] and *Botrytis cinerea* [41]. Collectively, these studies revealed that a diversity of protein architectures control incompatibility-driven cell death. Some domains and protein architectures are however shared by several *het* systems. The first domain found in common in different incompatibility genes is a cell death inducing domain termed HET (Pfam 06985) [34,42]. This domain shows a remote homology to TIR domains (Toll/interleukin-1 receptor) which control various aspects of immune defense and cell death in bacteria, plants and animals. Several TIR-domains display an NAD-hydrolase activity and lead to the production of NAD-derived secondary messengers [43–45]. In addition, several incompatibility genes encode NOD-like receptors (NLRs), a family of intracellular receptors with immune-related functions in animals, plants and bacteria [46–48]. Another domain common to several systems is a membrane targeting cell death execution domain termed HeLo that is related to the 4HB domain of the mammalian MLKL necroptosis executioner and to plant CC-domains controlling hypersensitive response [13,49]. Finally, several *het*-related genes encode gasdermines (GSDMs), a class of pore-forming cell death execution proteins with immune functions in animals and bacteria [32,33,50]. Many *het*-gene products thus have homologs with immune functions in other branches of the tree-of-life [51,52]. Along with the recent massive characterization of bacterial antiphage systems, work on fungal incompatibility contributes to illustrate trans-kingdom conservation of immunity and RCD mechanisms from microorganisms to metazoans and land plants [53]. Aside from these long-term evolutionary considerations, one pivotal question regarding fungal allorecognition systems centers on their evolutionary buildup within a particular species or taxon. How these genetic systems emerge remains only very partially understood. One hypothesis proposes that *het* genes originate from genes with pre-existing immune functions in xenorecognition (response to heterospecific non-self), subsequently co-opted for dedicated roles in conspecific recognition (allorecognition) within a given species or subclade. [14, 52, 54–56]. Put differently, genes originally functioning in the control and execution of immune RCD in the context of pathogen defense might have been refurbished (by exaptation) to trigger RCD in the context of conspecific non-self recognition.

Here, we set out to clone the *het-B* locus, the last of the nine originally described incompatibility loci in *P. anserina* that remained to be isolated. We find that although the system is genetically allelic, *het-B* incompatibility is determined by non-allelic interactions. *het-B* is a complex locus encompassing two adjacent genes: *Bh* encoding a HET-domain protein with a C-terminal predicted lectin fold domain and *Bp*, encoding a PII-like protein. These genes form highly divergent haplotypes: *Bh1-Bp1* and *Bh2-Bp2* in B1 and B2 strains with the respective allele products showing only ~60–70% identity. We find that incompatibility is solely determined by the interaction of BH1 and BP2 and that the lectin fold domain of BH1 determines recognition specificity. Population studies and genome analyses of closely-related species indicate that *het-B* haplotypes are under balancing selection and show trans-species polymorphism. The description of the full complement of *P. anserina het* genes now allows for a comparison of the genetic and mechanistic modalities leading to incompatibility in that species.

## Results

### *Pa_3_2660 (Bh1)* determines *het-B1* incompatibility

*Het-B* is an allelic system with two alleles, *het-B1* and *het-B2* originally identified in French isolate A and C (S1 Fig) [22]. Both alleles have been backcrossed into the *s* genetic background. Confrontation on solid medium of B1 and B2 strains leads to a barrage reaction (Figs 1A and S1). At the microscopic level, B1/B2 fusion cells undergo extensive vacuolization and cell lysis

(S1C Fig). Sexual crosses between B1 and B2 strains show normal fertility, *i.e.*, there is no sexual incompatibility associated to *het-B* (in contrast to *het-R/V* and *het-C/D/E* incompatibility) (S1 Fig).

*Het-B* was previously found to be located on chromosome 3 close to the centromere, linked to *het-s*, but on the opposite arm of the chromosome [57]. We thus mapped *het-B* relative to centromere linked markers located on that chromosome arm, namely *ura5*, *pex12*, and *Pa_3_3370* (Fig 1B). We measured 12 map units between gene *Pa_3_3370* and *het-B*, which yields an estimated position for *het-B* approximately 300 kb centromere proximal to *Pa_3_3370*. We inspected gene annotation in that region and noted that two genes (*Pa_3_2660* and *Pa_3_2610*, located respectively 295 kb and 305 kb centromere proximal to *Pa_3_3370*) encode proteins with HET domains (a cell death inducing domain common to several previously identified *het* genes) (S1 Table) (Fig 1B). We picked *Pa_3_2660* as first candidate gene because its HET domain is slightly more similar to those of *het-D, E*, and *R* (~45% identity instead of ~40% for *Pa_3_2610*). *Pa_3_2660* was deleted by gene replacement in a B1 strain and the *ΔPa_3_2660* strain was tested in barrage tests against B1 and B2 strains. Deletion of *Pa_3_2660* led to loss of the barrage reaction to B2 (Fig 1A). We conclude that *Pa_3_2660* participates in *het-B* incompatibility. The gene was renamed *Bh1*. Activity of *het* genes can be assayed in transformation assays with introduction of *het* alleles into incompatible backgrounds leading to reduced transformation efficiencies [14,32,34]. Introduction of *Bh1* into the B2 background led to reduced transformation efficiency, indicating that *Bh1* is necessary and sufficient to specify *het-B1* incompatibility (Fig 1C).

### *Pa_3_2650 (Bp2)* determines *het-B2* incompatibility

In order to explore the sequence polymorphism in that region now validated as being functionally involved in *het-B* incompatibility, we PCR-amplified the *het-B2* locus using 10 PCR primer-pairs, spanning a ~10 kb region around *Bh1* (Fig 1B). We found that primer pairs corresponding to an internal ~5 kb region encompassing *Pa_3_2630* to *_2660* did not amplify in the B2 background while primers in the flanking regions did. We chose a primer pair (primers *a* and *b*) in the conserved sequences flanking the predicted polymorphic region and obtained a 12 793 bp product in B2 (Figs 1B and 2), while the same primer pair yields an 8 877 bp product in B1. Comparison of the sequence of that region in B1 and B2 strains revealed an extensive polymorphism within ORFs *Pa_3_2630, 2640, 2650* and *2660/Bh* with nucleotide identities in the 70–80% range (Fig 2). In addition, the region showed several inversions, resulting in a different gene order for *Pa_3_2620, 2630* and *2640* and a different relative orientation of *Pa_3_2650* to *Bh*. Several intergenic regions, in particular between *Pa_3_2640* and *_2630* and *Pa_3_2620* and *2640*, appear specific to *het-B2* and are totally dissimilar between the two haplotypes. Closer inspection revealed that the novel sequence belongs to LTR retrotransposon insertions.

The simplest model for *het-B* incompatibility would be that incompatibility involves a single gene and results from allelic interactions between the two divergent alleles of *Bh/Pa_3_2660* (Fig 2). To test this model, we deleted *Bh2* (the *Bh* gene of B2 strains). Contrary to expectations, *ΔBh2* still produced a barrage reaction to B1 (Figs 3A, S2 and S3), thus ruling a simple model of allelic *Bh1/Bh2* interaction. *Het-B2* incompatibility must hence be controlled by a gene distinct from, but closely linked to *Bh2*. To identify this gene, we transformed B1 protoplasts with three B2 fragments (*ab, ac* and *db*) encompassing the different ORFs of the polymorphic region. Fragment *ab* contains all four polymorphic ORFs (*Pa_3_2640, 2630, 2650* and *2660*), *ac* only *Pa_3_2640* and *db Pa_3_2650* and *2660* (*Bh2*) (S2 Fig). Fragment *ab* and *db* yielded no transformants when introduced into the B1 background. Because deletion

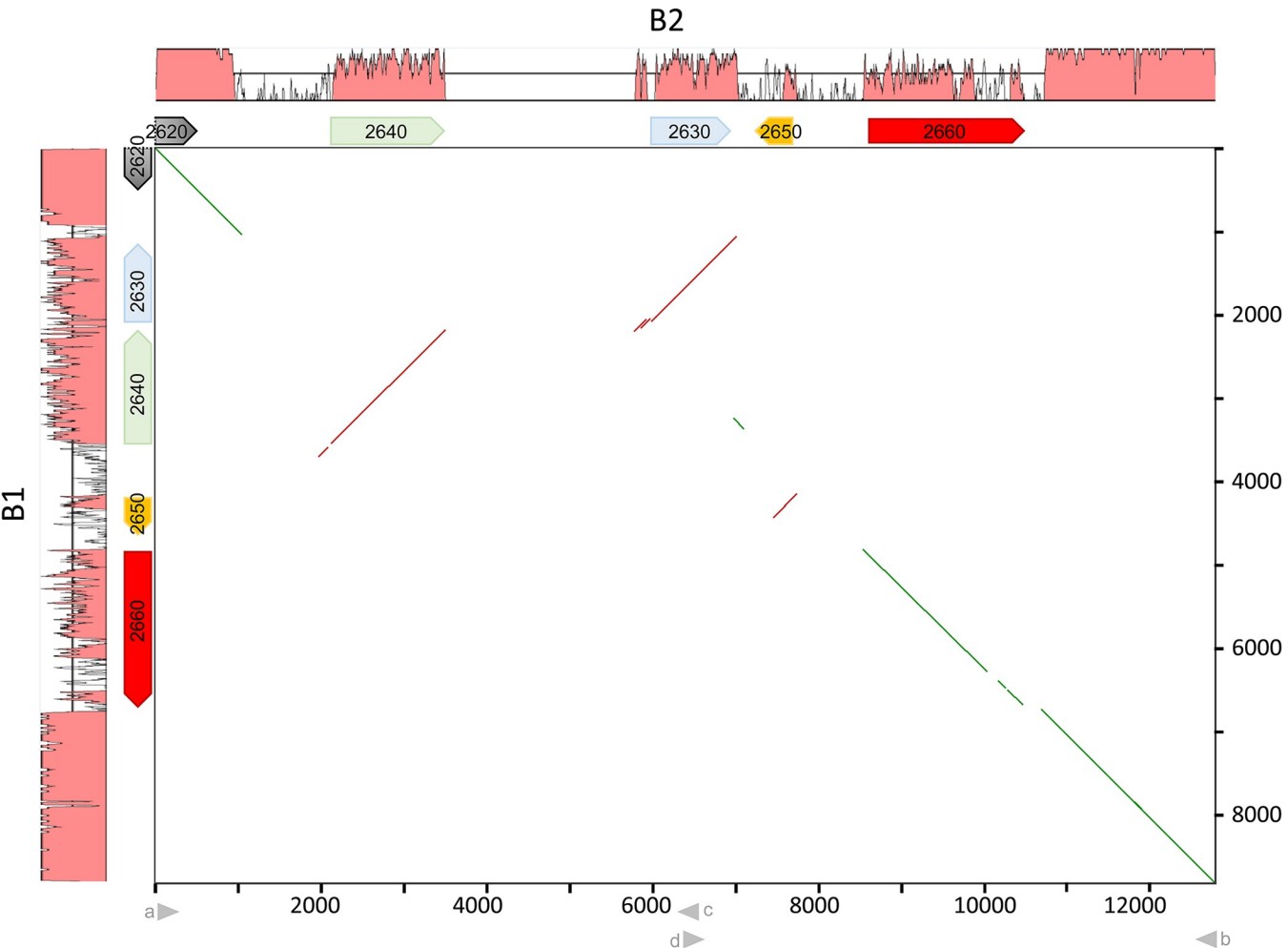

**Fig 2. Comparison of the *het-B1* and *het-B2* haplotypes.** Dot plot comparison of DNA sequences of the *het-B*-locus region in the B1 and B2 backgrounds. The B1 and B2 haplotypes correspond respectively to the vertical and the horizontal axes. The level of sequence similarity of each haplotype to the other is represented graphically. Numbering is in base pairs. In the ORF map shortened gene designations are used (e.g., 2660 is *Pa_3_2660*). In the dot plot, colinear and inverted regions are represented respectively as green or red lines. Positions of relevant oligos are given below the diagram as arrow heads.

of *Pa_3_2660* does not affect *het-B2* incompatibility (Figs 3A and S2), this observation suggests that the other ORF on fragment *db*, namely *Pa_3_2650* (that is the gene immediately adjacent to *Bh*) encodes the *het-B2* incompatibility factor. We deleted *Pa_3_2650* in the B2 background. A *ΔPa_3_2650 (B2)* strain shows no barrage reaction to B1 (Figs 3A, S2 and S3), indicating that *Pa_3_2650* is involved in *het-B2* incompatibility. *Pa_3_2650* encodes a PII-like protein, a type of small trimeric, metabolite sensing proteins described in bacteria [58]. *Pa_3_2650* was renamed *Bp*. We cloned the two alleles of *Bp*, *Bp1* and *Bp2*, and transformed them into B1, B2, *ΔBh1* and *ΔBh2* strains. As expected, introduction of *Bp2* in the B1 background led to reduced transformation efficiency (Fig 3B). In addition, *ΔBh1* strains transformed with *Bp2* produce a barrage reaction to B1 (S2 Fig). These results indicate that *Bp2* is necessary and sufficient to specify the *het-B2* incompatibility-type. We also deleted the *Bp1* allele in the B1 background and found that *ΔBp1* strains remain incompatible with B2 indicating that *Bp1* is not required for *het-B* incompatibility (Fig 3A).

Incompatibility interactions were assayed in all pairwise combinations between B1, B2, *ΔBp1*, *ΔBp2*, *ΔBh1*, and *ΔBh2* strains. Consistent with previous observations, *ΔBh1* and *ΔBp2*

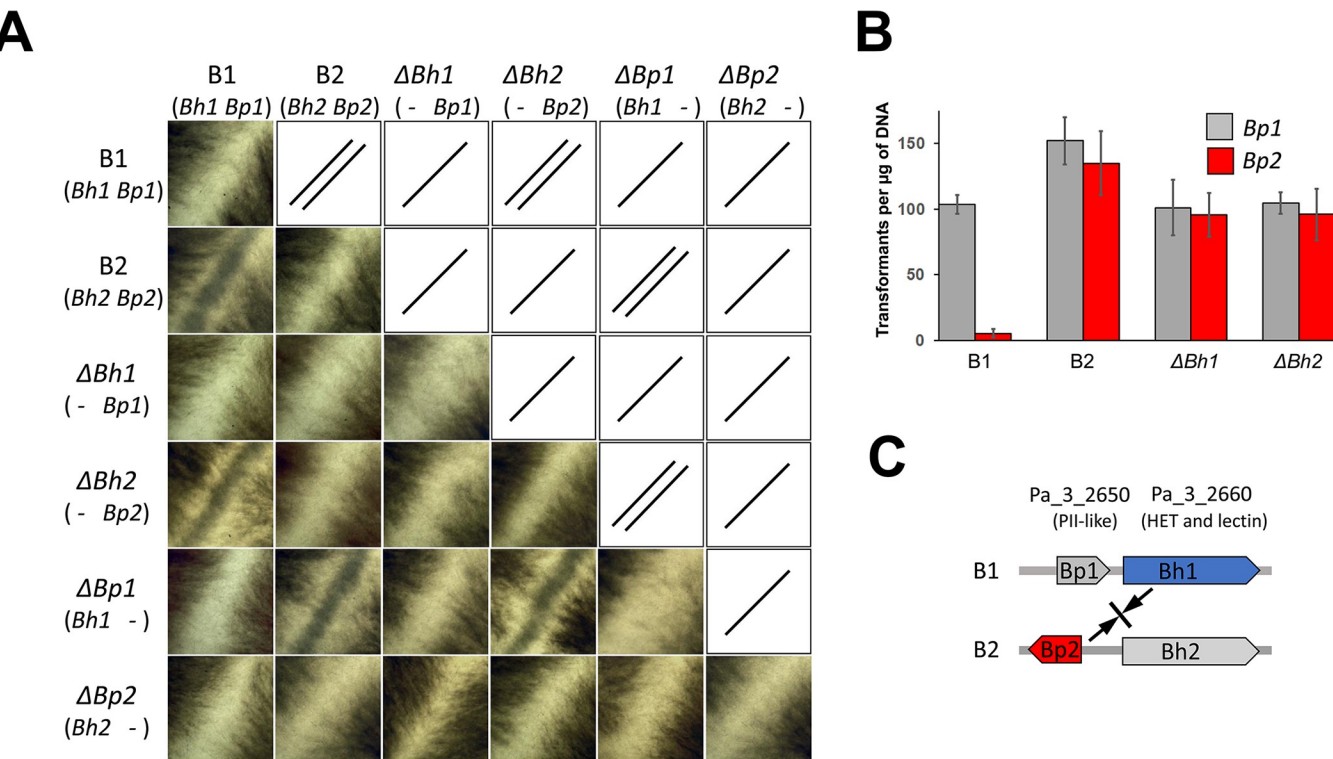

**Fig 3. *Bh1* and *Bp2* control *het-B* incompatibility.** (A) Strains as given were confronted in all combinations on solid corn meal agar medium and after 5 days at 26˚C the confrontation zone was imaged using a binocular lens (image size is 4.3x4.3 mm). Full test plates are given in S3 Fig. Under the strain designation, the constitution at *Bh* and *Bp* is recalled. On the top right part of the diagram, the barrage results are repeated graphically (/, for compatibility; //, for incompatibility) to improve readability. Note that *ΔBh1* and *ΔBp2* are compatible with all testers and that *ΔBh2* and *ΔBp1* retain the same incompatibility spectrum as the B1 and B2 strains from which they derive. (B) Transformation efficiency of the *Bp1* and *Bp2* alleles in different *het-B* genetic backgrounds. Note that *Bp2* leads to a strong reduction in transformation efficiency in the B1 background but not in *ΔBh1*. Results are quadruplicates with standard deviations. (C) Interpretative model of *het-B* incompatibility interactions, *het-B* incompatibility is asymmetric and determined solely by the *Bh1/Bp2* interaction, the reciprocal *Bh2/Bp1* interaction is compatible. *Het-B2* incompatibility type is determined by *Bp2* and *het-B1* incompatibility type by *Bh1*.

strains are both neutral in *het-B* incompatibility and compatible with all testers (Figs 3A and S3), while *ΔBp1* and *ΔBh2* have the same incompatibility spectrum as B1 and B2 respectively. These results as well as the result of the transformation efficiency experiments (Figs 1C and 3B) show that *het-B* incompatibility is determined by the pseudo-allelic *Bh1/Bp2* interaction. The reciprocal *Bh2/Bp1* interaction is compatible and neither *Bh2* nor *Bp1* are required for *het-B* incompatibility (Fig 3C). The minute physical distance between *Bh* and *Bp* as well as sequence divergence and rearrangements in that region presumably totally preclude recombination, hence *het-B* incompatibility appears genetically strictly allelic. This type of pseudo-allelic incompatibility is not unprecedented and has been described for *het-Z* [14].

### Annotation and comparison of *Bh* and *Bp* allele products

*Bh1* and *Bh2* both show a single exon and encode proteins of 622 and 638 residues respectively. The two allelic variants are highly dissimilar and display only 69% identity (Fig 4A). Three distinct domains are predicted. An N-terminal HET domain spans residue 1 to 240 (BH1 coordinates) (Fig 4A) and shares, as mentioned above, ~45% identity with the HET domain found in N-terminal position in the HET-D, E, and R NLRs. A predicted β-sheet-rich domain spans residue ~240 to 380. HMMER and Foldseek searches indicate that this domain is restricted to filamentous fungi. When present, it is nearly systematically found associated to HET (we term it

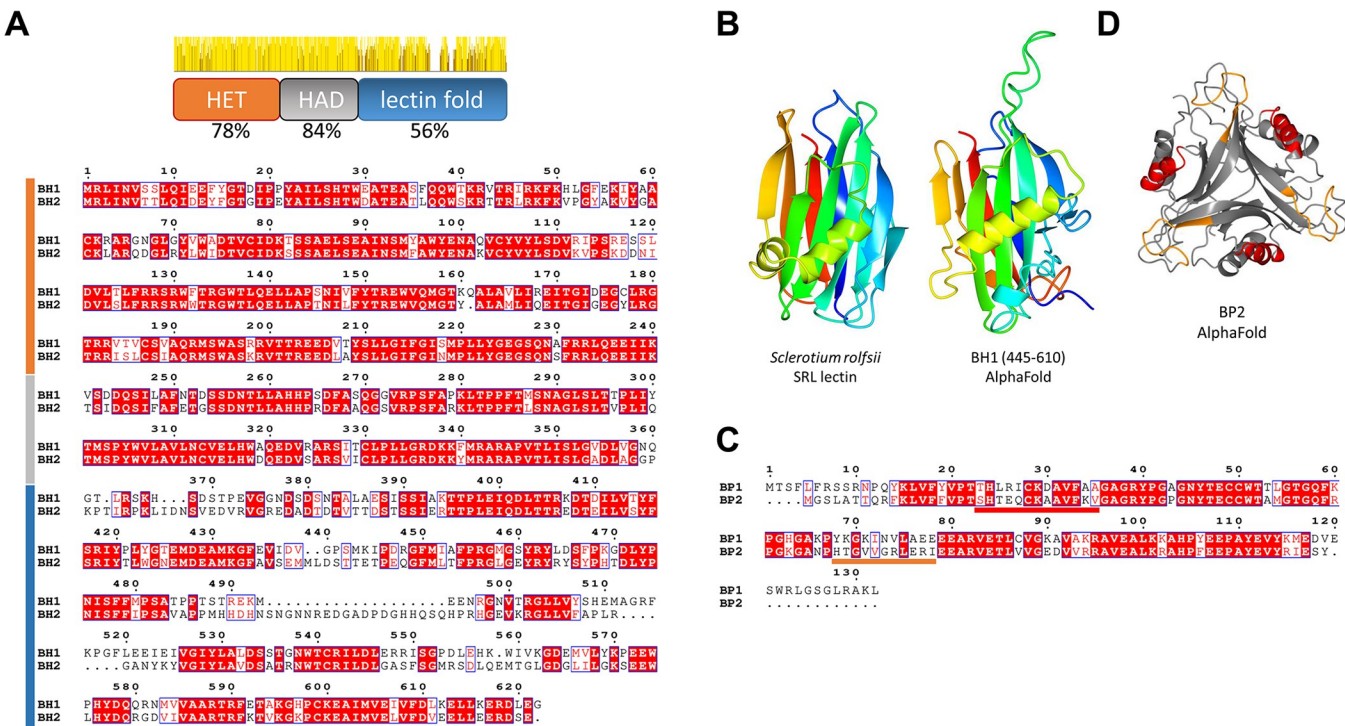

**Fig 4. Domain annotation and comparison of BH and BP proteins.** (A) Alignment of the BH1 and BH2 sequences. The diagram above the alignment gives the domain organization of BH and the level of identity between BH1 and BH2 in each domain together with a graphic representation of the sequence identity along the protein sequence. The position of the different domains are also given on the left side of the alignment with the same color-code as in the domain diagram. (B) An AlphaFold model of the BH1 C-terminal lectin fold domain is given together with the structure of the *Sclerotium rolfsii* SRL lectin (pdb:2OFC). (C) Alignment of the BP1 and BP2 sequences with the polymorphic T-loop and α1-regions colour code as in (D). (D) AlphaFold model of the BP2 trimer. The polymorphic T-loop and α1-regions are given in orange and red respectively.

HAD for HET associated domain). This domain is not found in NLR architecture HET-domain proteins. The C-terminal domain of BH1 (residue ~380 to 622) was identified as a lectin fold in Foldseek searches. This domain shows a predicted structural similarity to a family of fungal lectins, termed fungal fruit body lectins (FB-lectins) (Figs 4B and S4) [59, 60]. Fruiting-body lectins represent a subgroup of carbohydrate binding proteins with a fold related to invertebrate actinoporins [60,61]. These lectins are expressed in fruiting bodies in ascomycetes and basidiomycetes and were found to have a role in defense against parasites and micropredators [62–64]. The FB-lectin fold is a β-sandwich fold associated to a helix-loop-helix motif [59,60]. This C-terminal lectin fold domain is more divergent than the HET and HAD domains, with identity between BH1 and BH2 dropping below 60% in that region (Fig 4A). This HET-lectin fold association is not unique to *Bh* orthologs but is also found in 12 other gene products in *P. anserina* including *Pa_3_2610* mentioned previously and located in the same region as *Bh1*. In *N. crassa*, four HET-domain proteins show a lectin fold domain including NCU11383, which was found to be highly polymorphic and under balancing selection in *N. crassa* populations [17]. Globally in *Sordariales* 5–10% of HET-domain proteins display, like BH, a C-terminal lectin fold domain (13 out of 132 in *Podospora anserina*; 4 out of 65 in *N. crassa* OR74A).

*Bp1* and *Bp2* encode small proteins of respectively 132 and 115 residues, sharing only 62% identity (Fig 4C). *Bp* alleles display one predicted intron. BP proteins belong to the PII-like superfamily [58, 65]. PII-like proteins are small proteins (~100 in length), mainly studied in bacteria, forming triangular-shaped trimers and displaying a variety of sensing and signal

transduction roles [58,65]. PII-like proteins bind small metabolites in particular adenine nucleotides. Ligands are bound at each summit of the triangular trimeric assembly and binding involves a region called T-loop. AlphaFold and HSYMDOCK predict a PII-like trimeric fold for BP proteins (Figs 4 and S5). The T-loop region is divergent in sequence between BP1 and BP2. Outside of *Pezizomycotina*, the best hits to BP2 are PII-like proteins from gammaproteobacteria in particular from *Pseudomonales* (S5 Fig).

We analyzed the conservation of *Bh* and *Bp* in *Sordariales*. *Bp* is more conserved than *Bh*. *Bp* orthologs were found in the vast majority of species or strains while *Bh* orthologs were found in less than half (in respectively 71 and 33 of 83 analyzed taxa) (S2 Table). Most taxa lacking *Bp* orthologs (9/12) were thermophilic species. The tight clustering of *Bh* and *Bp* homologs was generally conserved (in 28 of 31 taxa presenting both *Bh* and *Bp* orthologs) although relative gene orientation varied (S2 Table). In the distantly related *Podospora didyma*, there are two PII-like paralogs [66]. As in *P. anserina*, one is adjacent to a *Bh* ortholog. The second, interestingly, is also adjacent to a gene encoding a HET-domain protein (otherwise unrelated to *Bh*). We analyzed genome neighbors of PII-like genes in *Pezizomycotina* and found 89 cases in which PII-like genes were immediately adjacent to HET-domain genes (this represents ~5% of PII-like homologs). These PII-like/HET gene pairs occurred in phylogenetically distant species including *Leotiomycetes*, *Dothideomycetes*, and *Eurotiomycetes* (S3 Table). The HET-domain genes linked to *Bp*/PII-like genes corresponds to at least three different lineages: the lineage typified by *Bh1* and two lineages lacking the lectin fold domain and instead displaying a domain with a conserved PSWSW-motif (also found in *N. crassa* TOL and PIN-C HET-domain proteins) or a variation thereof (PSWIPRW-motif). In one case (in *Rutstroemia* sp. NJR-2017), a PII-like gene was found adjacent to a NLR gene encoding a HET/NB-ARC/ANK protein. In two species (*Hyaloscypha* sp. PMI_1271 and *Cercophora newfieldiana*), a PII-like gene was flanked on both sides by HET-domain gene neighbors (belonging to distinct lineages).

## The C-terminal lectin domain of BH1 is responsible for allele specificity

For *het*-genes products with a NLR architecture (*het-E*, *D*, *R*, and *Z*) it has been found that the C-terminal variable region was responsible for allele specificity [14,30,31,67,68]. This prompted us to determine whether BH specificity could be determined by the more variable C-terminal lectin fold domain. We thus constructed a chimeric allele in which the C-terminal lectin fold domain of BH2 was replaced by the corresponding region from BH1. The chimeric *Bh2(Bh1/420-622)* allele was introduced in a *ΔB* background (lacking both *Bp1* and *Bh1*) and transformants were tested in barrage assays against B1 and B2. Transformants produced a barrage reaction to B2 indicating that the chimeric allele behaves as *Bh1* (Fig 5). We conclude that the allele specificity of *Bh* is determined by the lectin fold region. To determine whether the specificity region of *Bh* could be further restricted, we constructed two additional chimeric alleles specifically substituting highly polymorphic regions differing in BH1 and BH2, namely the region corresponding to a predicted α-helix located on the β-sandwich structure and a highly divergent loop connecting predicted β-strand 4 and 5 on the lectin fold (S4 Fig). The region encompassing the α-helix and strand 4 to 5 connection roughly corresponds to the sugar-binding region identified in FB-lectins [59]. Two chimeric alleles were generated: *Bh2 (Bh1/484-497; 547–575)* in which both regions are substituted by the BH1 sequence, and *Bh2 (Bh1/547-575)* in which only the α-helix region is substituted. When transformed into a *ΔB* background, none of the chimeras led to a barrage reaction to B2; in other words the exchanged regions do not lead to a switch to *Bh1*-specificity as observed for the *Bh2(Bh1/420-622)* chimera (S6 Fig). This result suggests that other allelic differences outside of the highly polymorphic exchanged regions are involved in *Bh* allele specificity.

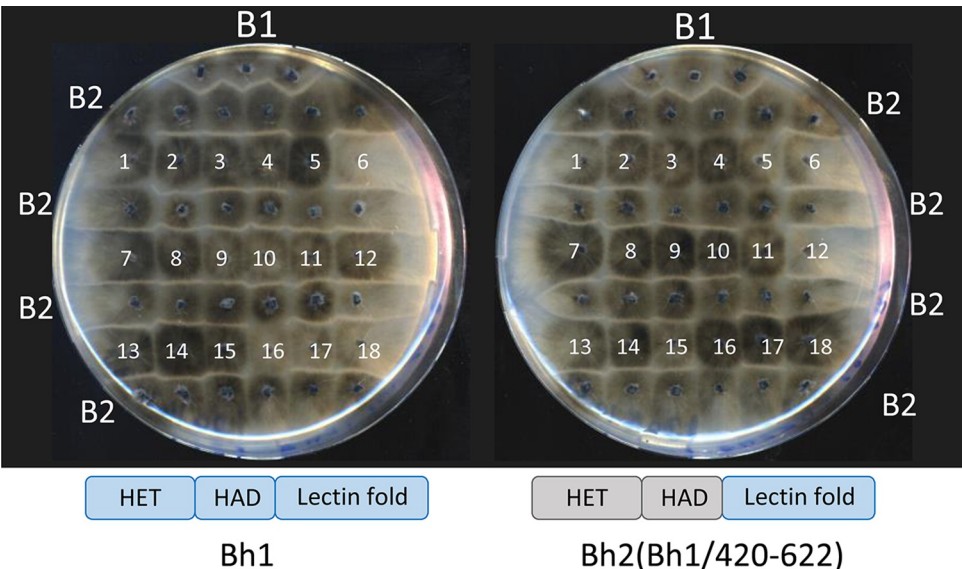

**Fig 5. The lectin fold domain defines *Bh*-recognition specificity in incompatibility.** *ΔB* strains (lacking the entire B1 haplotype) were transformed with either *Bh1* or the chimeric *Bh2(Bh1/420-622)* allele and individual transformants (as numbered) were assayed in barrage test in confrontation to B2. For *Bh1*, #2,3,4,6,7,8,9,10,11,12,13,14,15,17,18 produce a barrage reaction, for *Bh2(Bh1/420-622)*, #1,2,6,7,8,9,10,12,13,14,15,17 produce a barrage reaction. The upper line of the test is a B1 strain used as positive control. The domain diagram of *Bh1* and the chimeric *Bh2(Bh1/420-622)* allele are recalled.

A similar approach was also developed to analyze allele specificity of BP. Two regions of higher polymorphism were targeted, the so-called T-loop region connecting predicted strands β2 and β3 and the α1 helix (S6 Fig). Two chimeric alleles *Bp1(Bp2/20-31;64–74)* and *Bp(Bp2/64-74)* were generated and transformed into the *ΔB* background and assayed in barrage tests against B1. The chimeric alleles did not produce a barrage reaction to B1 suggesting as for *Bh*, that other differences outside of the polymorphic exchanged regions are involved in defining specificity.

## Evidence of balancing selection acting on *het-B*

Known *het* loci in *P. anserina*, *N. crassa*, and *C. parasitica* typically display signatures of balancing selection [12–17,69], such as balanced allele distributions, an excess of genetic diversity at linked positions, and the long-term maintenance of functional alleles through speciation events (trans-species polymorphism). We thus analyzed the *het-B* locus in *P. anserina* and closely related species in search of such signatures.

First, we determined the *het-B* genotype from previously published sequencing data of 106 *P. anserina* strains collected in Wageningen (the Netherlands) between 1991 and 2016 [13]. In addition, we genotyped by PCR two more strains available from 2016. For any given year, one or the other *het-B* alleles can dominate the population, but there is a tendency for *het-B1* to be the most frequent (Fig 6A, S4 Table). Overall in this collection, 69 strains were of the *het-B1* genotype and the remaining 39 had the *het-B2* genotype (S4 Table, Fig 6A), which deviates significantly from a perfectly balanced ratio of 1:1 (Chi-Square Goodness of fit $X^2_{df = 1}$ = 8.3333, $p$ = 0.0039). As individual years suffer from low sample sizes, we used PCR to genotype a more recent collection of 61 strains from the same locality that were sampled in the autumn of 2017 [13]. From these, 40 were *het-B1* and 21 were *het-B2*, mirroring the general ratio of 2:1 from previous years ($X^2_{df = 1}$ = 5.918, $p$ = 0.0150). Finally, the small French original *P. anserina*

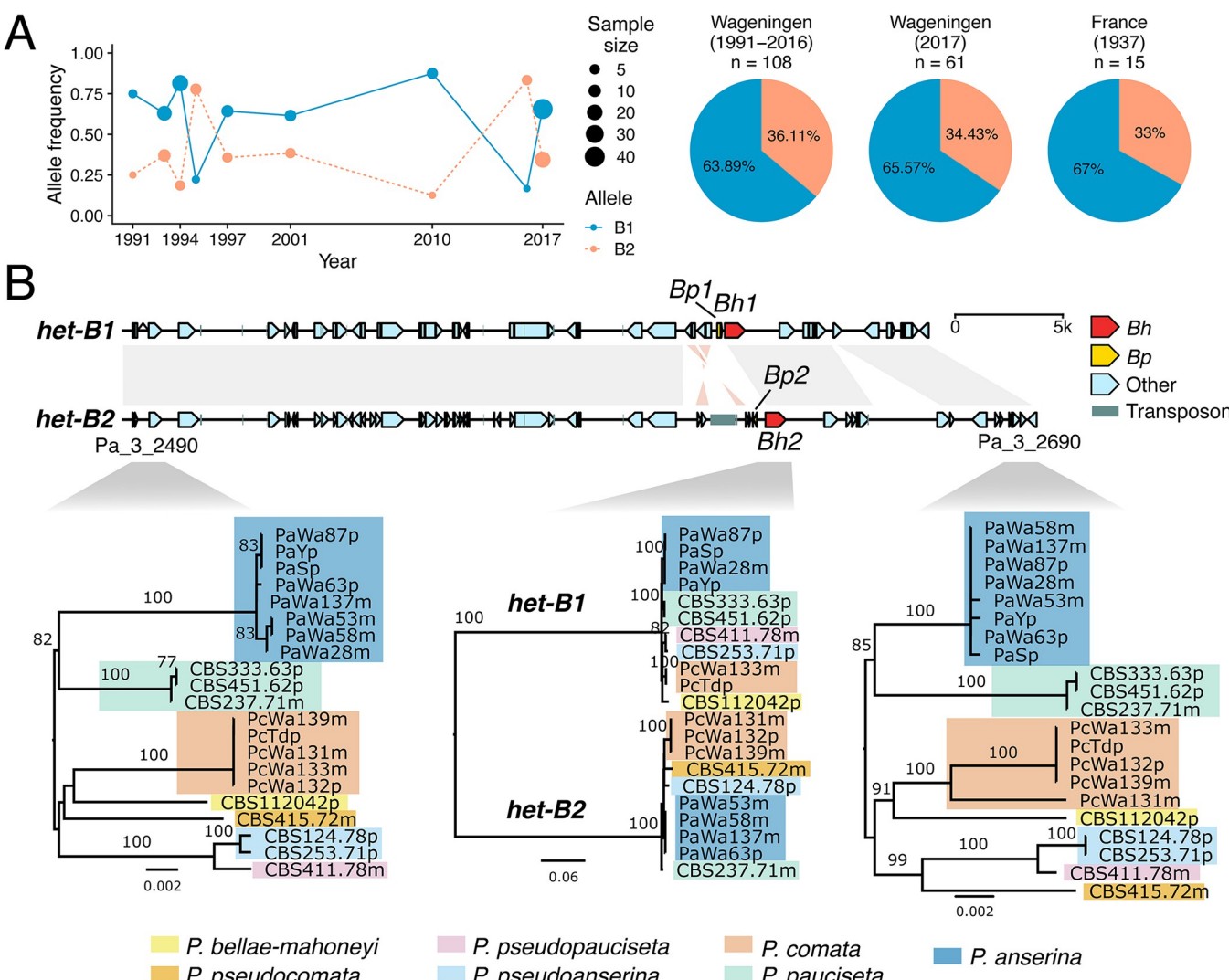

**Fig 6. Evidence of balancing selection on *het-B*.** (A) Allele frequency changes over time for the Wageningen Collection (the Netherlands), with a comparison of total *het-B* frequencies in three sample groups: a group of 108 strains from the Wageningen area (strains collected between 1991 to 2016 and data extracted from whole-genome sequencing), a second group of 61 strains from the same locality (collected in 2017 and genotyped by PCR) and a smaller collection of phenotyped French strains (for which sampling started in 1937). (B) Alignment of the *het-B* haplotypes and surrounding regions in *P. anserina*, along with maximum-likelihood phylogenies of *Bh* and unrelated genes at the flanks across the *P. anserina* species complex. Same-sense alignments are represented with gray links, while inverted alignments are in light red. Branch lengths drawn to scale as indicated by their scale bar (nucleotide substitutions per site). Branch support values are nonparametric bootstraps larger than 70%.

collection (for which sampling started in 1937) displayed a similar distribution (10 *het-B1* and 5 *het-B2*, $X^2_{df = 1}$ = 1.6667, *p* = 0.1967) (Fig 6A). Previous work demonstrated that the *P. anserina* strains actually comprise two reproductively isolated groups solely defined by the interaction of *het-V* and *het-R* [13]. However, these groups do not correlate with the segregation of other *het* genes [13]. For example, from the strains with whole-genome data, the B1:B2 ratio for *het-V* strains is 39:18, and for the *het-V1* strains is 30:19 (S4 Table), similar to the broader collection. We conclude from these data that allele frequencies seem balanced through time, but there is a persistent bias towards *het-B1*.

If the *het-B* locus currently evolves under balancing selection, as suggested by the observed allele frequencies, then we should detect an excess of genetic diversity at linked positions [70].

Hence, we screened published estimates of pairwise nucleotide diversity ($\pi$), Watterson's theta $\theta_W$, and the Tajima's D statistic [13] in the area surrounding the *het-B* locus. As expected, the *het-B* locus directly locates in a narrow window of high genetic diversity and positive Tajima's D, similar to the *het-S* and *het-C* loci located on the opposite arm of the same chromosome (S7A Fig).

Finally, we inspected the *het-B* sequence in the recently characterized genomes of strains belonging to the *P. anserina* species complex [71]. Phylogenetic analyses revealed that the topology of the *het-B* haplotype (represented by *Bh*) follows the split of the two alleles, *Bh1* and *Bh2*, for all species where at least two strains are available (Fig 6B). By contrast, flanking genes return the expected species phylogeny. Moreover, the general architecture of the two haplotypes is conserved across the species complex, further supporting that it predates all speciation events (S7B Fig). Apparently, the B1 and B2-haplotypes can be traced a bit further down the phylogeny as *Bh1-Bp1* and *Bh2-Bp2*-type alleles can also be found respectively in *Podospora setosa* and *Cercophora samala* which are more distantly related to *P. anserina* (S7C Fig) [66].

## Discussion

The genetic characterization of incompatibility genes in *Podospora* by Georges Rizet and colleagues started in the 1950's in parallel with the studies on incompatibility in Neurospora pioneered in particular by Laura Garnjobst [72–75]. Jean Bernet described a total of nine *het* loci by analyzing a collection of 15 strains isolated from various regions in France [22]. The *het-B* locus is the last of these originally identified *het* loci that remained to be cloned. We have isolated *het-B* by a combination of positional cloning and candidate gene approaches. *Het-B* is an allelic system but we find that incompatibility in fact involves non-allelic interactions between two distinct genes *Bh* and *Bp* which reside in the same locus.

### The HET domain, a central player in fungal RCD

*Bh1* encodes a protein with a HET domain [34]. HET domains appear once again as central players in fungal RCD. We now know that four out of nine *het* genes in *Podospora* present this domain (*Bh1*, *het-D*, *E* and *R*). It is is also found in four *het* systems in *N. crassa* (*tol*, *het-6*, *het-E* and *pin-c*) and three in *C. parasitica* (*vic1*, *6* and *7*) [52]. However, it is not associated to any of the five *het* systems identified in *A. fumigatus* [16]. While *het-D*, *E*, and *R* are paralogs with an NLR architecture, in *het-B*, the HET domain occurs in a non-NLR context, associated to a C-terminal lectin fold domain. This domain architecture is not restricted to *Bh* but appears reasonably common with roughly 5–10% of the HET-domain proteins in *Sordariales* sharing this arrangement. By analogy with other HET-domain incompatibility systems it appears likely that the HET domain of BH1 is responsible for RCD execution [42]. How HET-domains induce cell death is currently unknown. TIR domains function in defense and RCD through NAD hydrolysis and production of NAD-derived secondary messenger molecules [44]. The remote homology between TIR and HET, and their shared occurrence in NLR architectures might suggest a related mode of action [43]. It is likely that the biological function of HET domains reaches beyond incompatibility. Indeed, only a minute fraction of the HET domains in *P. anserina* occurs on *het*-gene products (of 132 HET-domain genes only four are incompatibility genes) and apparently the same is true in other species like *C. parasitica* and *N. crassa*, with so far respectively 3 out of 94 and 4 out of 65 HET-domain genes identified as allorecognition-related genes. The HET domain is one of the most frequent domains in fungal genomes. In *Sordariales* for instance, it is the fourth most abundant Pfam annotated domain (after Major Facilitator Superfamily, Protein Kinase and Fungal Zn [2]-Cys [6] binuclear cluster) with on average 117 HET domains per strain or species. Strikingly, HET is the Pfam-

annotated domain showing the greatest variation in copy number between taxa (S5 Table). Considering abundance and variability of HET-domain genes, the broader understanding of the function of this domain in fungal biology, beyond its role in incompatibility, remains a significant gap in current knowledge. Recently it was reported that in *N. crassa*, lineage specific gene clusters with HET-domain genes but the functional significance of this genomic association in not well understood [76].

## A lectin fold determines recognition of a PII-like protein

In NLR-based *het* systems, it is proposed that activation of cell death execution domains involves recognition and binding of the partner protein by the C-terminal variable region followed by activation of the execution domain presumably by an oligomerisation process [14,67]. Based on the construction of a chimeric allele, it appears that the lectin fold region of BH is responsible for the ligand recognition step that is recognition and binding of BP2. Biochemical approaches are required to establish that incompatibility involves a direct BH1/BP2 interaction, but this model is at present the most parsimonious.

Fold prediction and comparisons suggest that the C-terminal domain of BH has a lectin fold. Lectins are proteins that bind glycans with high specificity. Interestingly, these highly specific sugar-binding abilities entitle lectins to play a variety of roles in the control of biotic interactions in immunity and symbiosis in different phyla [64,77–79]. The term lectin covers a wide structural diversity [80,81] and the lectin fold detected in BH is distantly related to a lectin family known as FB-lectins, structurally similar to invertebrate actinoporins [61] [82],). This lectin domain thus adds to the list of domains that function in incompatibility and have roles in immunity in other phyla. The putative binding partner of BH is the PII-like BP protein. PII-like proteins form a large superfamily conserved both in pro- and eukaryotes [58,65]. Functional characterization of several prokaryotic PII-like proteins indicates that these proteins function as sensors of small metabolites, in particular adenine nucleotides. The trimeric PII-like fold leads to the formation of three binding sites involving in particular a loop region termed T-loop that participates in the formation of a ligand binding pocket. PII-like proteins can sense cues of metabolic state, for instance ATP/ADP ratios. Small ligand binding allosterically modifies their trimeric structure and subsequent binding to downstream partners. Exceptionally, enzymatic activity can also be associated to PII-like proteins, but in the general case they are thought of as sensors of metabolites [83]. Despite their vast conservation in *Pezizomycotina*, the biological role of PII-like proteins in fungi is unknown. BP2 might represent the first fungal PII-like protein to which a biological function is associated. Outside of *Pezizomycotina*, the closest homologs to fungal BP homologs are bacterial PII-like proteins from *Pseudomonales*.

It has not been possible to delimit precisely the protein regions determining BH and BP specificity in incompatibility. Presumably, the very high divergence between the allele products hindered our approach based on construction of chimeric alleles, a strategy that was reasonably successful in delimiting specificity regions in other less divergent *het* genes like *het-C* and *het-S* [84,85]. It remains that the predicted small ligand binding site of both BH and BP are among the most variable parts of the protein. Considering that both lectins and PII-like proteins are binders of small molecule moieties, one could imagine that the BH/BP interaction involves recognition of a metabolite that would be bound in "sandwich" simultaneously by the T-loop region and the carbohydrate binding region of BH. The specificity difference between BP1/BP2 and BH1/BH2 could then either be related to the inability of the partners to bind that specific metabolite or else be due to the fact that partner recognition simultaneously involves recognition of the metabolite and of a protein-protein interaction surface. Alternatively,

recognition might be unrelated to the presumed metabolite binding ability and only involve interaction of the two proteins without assistance of a small molecule ligand.

## Trans-species polymorphism and origin of the *het-B* incompatibility system

The *het-B* locus shows trans-species polymorphism, that is, several species closely related to *P. anserina* also show the *het-B1/het-B2* polymorphism making it likely that *het-B* incompatibility occurs also in these species. Apparently, *het-B* polymorphism is ancient and has been retained for long periods, across several speciation events. It remains possible that introduction of *het-B* alleles in populations of given species of the *Podospora* species complex occurred by introgression. Adaptive introgression of incompatibility alleles has been documented in *Ophiostoma ulmi* [86]. This scenario is however unlikely since it would require multiple independent events early in the divergence of the species complex, judging by the accumulation of differences in *het-B* haplotypes. At any rate, there is evidence that selection is acting to maintain polymorphism at *het-B*. At a macroevolutionary scale, the *Bh-Bp* gene pair is conserved in species where there is apparently no *het-B* polymorphism. In *N. crassa*, the *Bh* and *Bp* homologs (NCU04913 and NCU04912) do not appear to function in heterokaryon incompatibility since they are not strongly polymorphic in populations and show no mark of balancing selection [17]. This observation suggests that the *Bp/Bh* gene pair has a function that predates emergence of incompatibility. In *P. anserina*, only the *Bh1/Bp2* gene pair functions in incompatibility, while *Bh2* and *Bp1* play no role in the process. The two genes inactive in incompatibility, *Bh2* and *Bp1*, are however not pseudogenized in the *Podospora* species complex, also arguing for an additional function distinct from heterokaryon incompatibility. The genome vicinity of *Bp* (PII-like) and HET-domain genes appears conserved in many *Sordariales* where it could either by the results of the general micro or mesosynteny [87] or reflect a possible functional importance of this genomic association. The fact that HET-domain genes (some of which are not paralagous to *Bh*) can be found adjacent to PII-like genes in other species very distant from *Sordariales* favors the latter hypothesis. It is possible that the *Bh/Bp* association reflects a functional coupling as described for other *het*-gene related partners [13,14,32,49,67]. It has been proposed that incompatibility systems might derive from pre-existing cellular systems controlling immune RCD in the context of pathogen defense [52,54,56,88]. How does *het-B* gene complex locus fit into this exaptation hypothesis? Although this hypothesis is at present highly speculative, by analogy with other *het* systems, it could be envisioned that the *Bh-Bp* gene pair originally functions in xenorecognition. It is conceivable that the BP PII-like recognizes a small molecule (possibly an adenine nucleotide) acting as a cue of pathogenic challenge and consequently interacts with BH to trigger HET-mediated RCD. Considering the centrality of nucleotide-derived signaling in immune pathways [89,90], the involvement of a PII-like protein in this allorecognition and RCD system calls for further exploration of its sensing specificity and biological function.

Presence of trans-species polymorphism, high Tajima's D values, and high frequency of both allele types in different population samples all suggest that *het-B* haplotypes are under balancing selection. Balancing selection appears to be a property shared by all incompatibility systems in Podospora [13,23,91] and beyond by *het* systems in *Pezizomycota* in general [12,14–17,69,92]. Yet, rather than being strictly equilibrated, allele frequency at *het-B* are slightly skewed with a 2:1 (B1:B2) ratio. This observation suggests that allele frequency is not solely determined by balancing selection on the allorecognition function. At *het-S*, allele frequencies were also found to deviate slightly from equilibrium [91]. In that specific case, it was proposed that meiotic drive could be responsible for skewing *het-s/het-S* allele ratios in favor of *het-s* (the

driver allele), an argument that cannot be made for *het-B*. One hypothesis to explain the skewed ratio is that possible xeno- and allorecognition functions overlap in *Bh-Bp* and that there is some fitness advantage associated to the *het-B1* haplotype in the xenorecognition function. In other words, this hypothesis poses that *het-B1* does slightly better than *het-B2* in an hypothetical immune function distinct from incompatibility. An alternate hypothesis is that linked polymorphisms (for instance in *Pa_3_2640 and/or 2630*) entail a slight fitness difference, skewing allele ratios through hitchhiking.

## Genetic architectures of allorecognition in Podospora

Now that all known incompatibility systems in *P. anserina* are characterized, at least for their basic structural and mechanistic properties, it is possible to discuss the different genetic modalities that underlie allorecognition in that species. Fig 7 recapitulates the different genetic topologies encountered for Podospora *het* sytems. *Het-B* was defined as genetically allelic but in fact involves non-allelic, or rather pseudo-allelic interactions. It becomes apparent that although the majority of the systems are defined as allelic, among them only *het-s* is a true allelic system, involving interaction of two allelic variants of the same gene (we term it euallelic). The four other allelic systems are either pseudo-allelic or idiomorphic. Pseudo-allelic interactions can be reciprocal (for *het-Z*) [14] or non-reciprocal (for *het-B*). It is of note however, that the *het-Z* system is also non-reciprocal (asymmetric) with respect to sexual incompatibility (S1 Fig). The *het-Q* system is idiomorphic, as in the *N. crassa* mating-type associated incompatibility [32,37], while in the case of *het-V* one encounters a most peculiar situation where incompatibility involves a presence/absence polymorphism of two genes, *het-Va* and *het-Vb* [13].

Pseudo-allelic systems could have evolved in two possible scenarios. First, they could initially emerge as non-allelic systems and secondarily, rearrangements creating linkage between partner genes could be selected because tight linkage can prevent postzygotic deleterious effects (such as the formation of self-incompatible progeny) in the event of outcrossing [26]. The Neurospora *het-c/pin-c* allorecognition locus conforms to such a scenario as it has evolved as a consequence of a rearrangement event whereby the *pin-c* HET-domain gene was inserted near the *het-c* locus [92]. This hypothesis can also explain the convergent linkage of distinct NLR-encoding genes to *sec9* SNARE homologs at allorecognition loci in *Botrytis*, *Cryphonectria*, *Neurospora*, and *Podospora* [14,39,41]. Alternatively, the partner genes evolving into *het-*

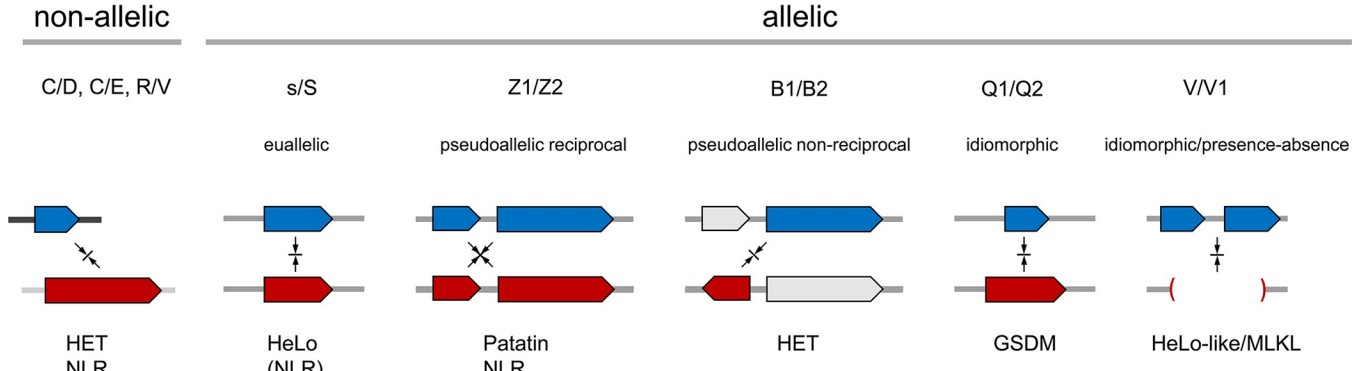

**Fig 7. Comparison of the genetic and domain architectures of the different incompatibility systems in *Podospora anserina*.** Diagram illustrates the different genetic architectures underlying incompatibility. Genetically, systems are categorized into non-allelic and allelic systems. Antagonistic incompatibility partners are represented in blue and red. For each system, the type of RCD execution domain is given (HET, HeLo, Patatin, GSDM) while systems involving NLRs are marked. For s/S, the system involves an NLR (NWD2) but this NLR is not directly involved in the incompatibility reaction (hence the parentheses). See text for details and references.

alleles might have already been linked at the time of emergence of the incompatibility system (as a result of a pre-existing functional coupling). This scenario for instance can explain the idiomorphic nature of the *het-Q* system that likely evolved from a protease/gasdermine gene cluster [32]. For *Bh/Bp*, we favor this hypothesis of pseudo-allelism resulting from pre-existing linkage between the partner genes because the genes are already paired in *N. crassa* where the locus apparently does not function as a *het* locus.

A question remains as to the existence of additional *het*-genes in *Podospora anserina* possibly not represented in the initial set of 15 French strains analyzed genetically [22]. Studies in *C. parasitica* suggest that different populations of that species display different sets of *het* genes [93]. Analysis of other strain sets and populations is now necessary to determine whether or not other *het* systems exist in that species.

The pioneering genetic work of G. Rizet, K. Esser and J. Bernet have established *P. anserina* as a model system to study fungal incompatibility [22,73,94]. Cloning of the originally identified *het* genes is now completed and has led to some interesting insights into prion biology, fungal RCD, meiotic drive, and reproductive isolation [13,25,52,95]. Nevertheless, many key questions remain, in particular regarding the modalities of cell death execution, the exact adaptive value of these systems, and most intriguingly, the steps leading to their evolutionary buildup.

## Material and methods

### Strains and media

*P. anserina* strains used in this study were wild-type *het-B1* and *het-B2* strains [22] and the *Δku70* strain for gene inactivation experiments [96]. All strains were cultivated at 26˚C on D0 corn meal medium for barrage assays and M2 for crossing [97]. Protoplast transformation was performed as previously described [97,98].

### Genetic mapping

The *het-B* locus was mapped at 13 map units from the centromere of chromosome 3 on the arm opposite to *het-s* (*het-B* shows 13% second division segregation) [57]. *Het-B* was further mapped with respect to three genetic markers located on the same chromosome arm, *ura5* (Pa_3_4280) [99], *pex12* (*Pa_3_3790*::*nat*) [100] and *Pa_3_3370*::*hph*. Crosses in which *het-B* and these individual markers segregated yielded recombination rates between *het-B* and respectively *ura5*, *pex12* and *Pa_3_3370* of 35% (31/60 analyzed spores), 17% (19/109 analyzed spores) and 12% (7/56 analyzed spores). With an estimated 25 kb per 1% recombination rate, the mapping yields an estimated position of *het-B* approximately 300 kb (12x25kb) centromere proximal to *Pa_3_3370* and approximately 325 kb from the centromere. The actual physical distance of *Pa_3_2660* (*Bh*) to *Pa_3_3370* is 296 kb.

### Gene inactivations and plasmid constructs

For inactivation of *Pa_3_2660* in the B1 background (*Bh1*), a fragment upstream of the ATG codon was amplified with oligos occ551 and occ552 and a fragment downstream of the stop codon was amplified with oligos occ551 and occ554 using genomic DNA of the *het-B1* strain as template (oligos are listed in S6 Table). Both fragments were inserted in the orientation collinear with the genome sequence in SKhph, a pBlueScript II-derived vector containing the *hph* gene inserted in the polylinker region. The resulting plasmid was digested with *Nru*I and *Bgl*II and 5 μg of the 3.6 kb *Pa_3_2660*::*hph* deletion cassette was used to transform a *ΔKu70 het-B1* strain [96]. Hygromycin-resistant transformants were selected, screened by PCR and

backcrossed to the *het-B1* strain to eliminate the *ΔKu70* mutation. Inactivation of *Pa_3_2660* in the B2 background (*Bh2*) was performed similarly using a fragment upstream of the ATG amplified with oligos occ608 and occ609 and a fragment downstream of the stop codon amplified with oligos occ610 and occ611. The resulting *Pa_ 3_2660(B2)::hph* deletion cassette was used to transform a *ΔKu70 het-B2* strain. Inactivation of *Pa_3_2650* in the B1 background (*Bp1*) was performed in the same way, the upstream and downstream flanks were amplified respectively with oligo pairs occ1182/occ1183 and occ1184/occ1185. For inactivation of *Bp2*, the oligo pairs for amplifying the upstream and downstream flanks were occ637/occ638 and occ639/occ640 respectively. For the inactivation of the whole *het-B1* haplotype the same procedure was repeated using as flanks a fragment downstream of the *Pa_3_2660* ORF amplified with oligo pair occ553/occ554 and a fragment upstream of the *Pa_3_2650* ORF amplified with oligo pair occ1182/occ1183.

For cloning of *Bh1* (*Pa_3_2660*), *Bh1* was amplified with oligos occ620 and occ621 with Q5 DNA polymerase (NEB, Ipswich MA) using genomic DNA of a *het-B1* strain and cloned into the p1 vector, a pBlueScript-II derived vector containing the *nat1* nourseothricin acetyl transferase gene (the resulting plasmid is termed p1-BH1L). *Bh2* was amplified similarly with oligos occ622 and occ623 and using genomic DNA of a *het-B2* strain (p1-BH2). *Bp1* and *Bp2* (*Pa_3_2650*) were amplified with oligos occ636/occ552 and occ634/occ635 respectively and cloned into the p1 vector (respectively p1-BP1 and p1-BP2).

The *Bh2(Bh1/420-622)* chimeric allele was constructed by PCR-driven overlap extension [101] using oligos osd120 and occ621 to amplify the 5' part of *Bh1* using the p1-BH1L plasmid as template and oligos osd121 and occ622 to amplify the 3' part of *Bh2* using the p1-BH2 plasmid as template. The resulting reconstituted PCR fragment was cloned using the Zero Blunt TOPO PCR cloning kit (Invitrogen, Carlsbad CA) and subcloned into the p1 vector as a *Spe*I-*Not*I restriction fragment. The *Bh2(Bh1/547-575)* chimeric allele was generated using the Q5 Site-Directed Mutagenesis kit (NEB, Ipswich MA) using oligos osd126 and osd127 and p1-BH2 plasmid DNA as template. *Bh2(Bh1/484-497; 547–575)* was generated similarly using oligos osd140 and osd141 and using p1-BH2(BH1/547-575) plasmid DNA as template. The *Bp2(Bp1/64-74)* chimeric allele was generated similarly using oligos osd128 and osd129 and using p1-BP2 as template. The *Bp2(Bp1/20-31;64–74)* chimeric allele was generated similarly using oligos osd138 and osd139 and using p1-BP2(BP1/64-74) as template.

## PCR mapping of the *het-B* polymorphic region

Genomic DNA from *het-B1* and *het-B2* strains was used as template with 10 primer pairs spanning the tentative *het-B* region. Primer pairs from the centromere proximal to centromere distal direction, were occ572/occ573, occ566/occ567, occ564/occ565, occ574/occ575, occ576/occ577, occ568/occ569, occ551/occ552, occ553/occ554, occ578/occ579 and occ570/occ571. The sequence of the B2 haplotype was deposited in Genbank (accession number: OR910067).

## PCR screening of *het-B* genotype

Strains of *P. anserina* collected in the Wageningen area in 2016 and 2017 were screened for their *het-B* genotype by PCR using genomic DNA as template and using three oligos simultaneously: occ611 (common to *het-B1* and *het-B2* and located dowstream of *Bh*), occ636 (specific to *het-B1* and located upstream of *Bp1*) and occ635 (specific to *het-B2* and located upstream of *Bp2*). This three primer PCR generates a 3868 bp product (occ611/636) in *het-B1* strains and 4454 bp product (occ611/635) in *het-B2* strains.

## Microscopy

*P. anserina* hyphae were imaged on solid medium with a Leica DMRXA microscope equipped with a Micromax charge-coupled device (CCD) (Princeton Instruments) controlled by Meta-Morph 5.06 software (Roper Scientific). The microscope was fitted with a Leica PL APO 63x immersion lens. To analyze incompatible anastomoses, a *het-B1* strain expressing GFP was confronted to a *het-B2* strain expressing RFP. A *het-B1* strain expressing RFP was used as control for compatible anastomoses.

## Sequence analyses and structure prediction

Protein sequence alignments were performed with Clustal Omega (www.ebi.ac.uk) and graphically displayed in jalview and with ESPript (esprit.ibcp.fr). For Dot plot alignments YASS (bioinfo.univ-lille.fr/yass/yass.php) and VISTA (genome.lbl.gov/vista) were used. Protein folds were predicted using Alphafold using Colabfold [102] and visualized with the CCP4 Molecular Graphic Program. Trimeric fold prediction for BP were performed using HSYMDOCK (huanglab.phys.hust.edu.cn/hsymdock/) and AlphaFold. Fold comparisons were performed with FoldSeek (search.foldseek.com/search) and DALI (ekhidna2.biocenter.helsinki.fi/dali/). Sequence logos for BP and BH homologs were generated using HMMER (hmmer.org/). Gene coordinates for *Bh* and *Bp* homologs from *Sordariales* as well as Pfam-annotations were recovered at Mycocosm (mycocosm.jgi.doe.gov/mycocosm/home). Gene accession numbers and sequences of BP homologs and HET-domain proteins in *Pezizomycota* were recovered using psi-BLAST at ncbi (ncbi.nlm.nih.gov/).

## Genomic analyses

Strains of *P. anserina* have been collected around the city of Wageningen (the Netherlands) since 1991, allowing for a characterization of *het* gene allele frequencies in nature. The genomes of most strains in the collection sample until 2016 have been previously sequenced with Illumina HiSeq paired-end technology [13]. Likewise, whole genome assemblies have been published for the seven known members of the *P. anserina* species complex [71,103–105]. We assembled the short-read data with SPAdes v. 3.12.0 [106], using the k-mers 21, 33, 55, 77 and the *--careful* option as in Ament-Velásquez et al. (2023) [71]. We used BLAST suit 2.15.0 [107] and the script *query2haplotype.py* v. 1.92 to find and retrieve the *het-B* locus and neighboring genes (available at https://github.com/SLAment/Genomics). Annotations were obtained from Ament-Velásquez et al. (2023) [71] and adjusted to match the *het-B* haplotypes with *GFFSlicer.py* v. 4.2 in the same GitHub repository. Haplotype alignments were performed with Minimap2 v. 2.26 [108] with parameters -X -N 50 -p 0.1 -c -B 2. Synteny plots were produced with the gggenomes v. 0.9.12.900 package [109]. The Chi-Square Goodness of fit tests were calculated with the *chisq.test* function in base Rv. 4.1.1. The annotation of the B2 allele of *Pa_3_2660* (Bh2) contained an intron in some species. However, inspection of RNA-seq data for the *P. comata* strain Wa131- (*het-B2*) confirmed there is no evidence for such intron and hence it was removed from the annotations.

To investigate trans-species polymorphism, we extracted the nucleotide sequence of the genes *Pa_3_2490* (plus 200bp on each side), Pa_3_2660/*Bh*, and Pa_3_2690 (plus 200bp on each side) from the assemblies of the *P. anserina* species complex [71]. Multiple sequence alignments were produced with the default options of the online version of MAFFT [110], last consulted on 2023-12-07 (https://mafft.cbrc.jp/alignment/server/). Maximum-likelihood phylogenies were inferred with IQ-TREE v. 2.2.2.9 [111] with parameters *-m MFP -seed 1234 -b 100*.

## Supporting information

**S1 Fig. Incompatibility systems in *Podospora*, and macro- and microscopic manifestations of *het-B* incompatibility.** (A) The different genetically defined *het* systems of *P. anserina* are shown. Allelic systems are given in blue and non-allelic systems in red. *het-V* is involved both in allelic and non-allelic interactions and is given in purple. Opposing arrows represent an incompatible interaction. Full arrowheads indicate normal fertility, open arrowheads indicate sexual incompatibility. The arrow direction represents the direction of the cross, with the arrow pointing from the male parent to the female parent in the cross (for example for the V/V1 interaction, the diagram denotes that a male V1 x V female cross shows sexual incompatibility while the opposite cross is fertile). In the s/S interaction, the dashed arrowhead denotes the spore-killing reaction occurring in S male x s female crosses and leading to specific abortion of S spores. **B.** Barrage reaction (incompatibility) but lack of sexual incompatibility between B1 and B2. The strains of the given genotypes were confronted on corn meal agar and grown for 6 days in the dark (upper panel). The same plate was imaged again after a week under constant illumination (lower panel). Full incompatibility genotypes for relevant loci are as follows: B1: *het-B1*, *het-c2*, *het-d3*, *het-e4*, *het-r*, *het-V;* B2: *het-B2*, *het-c2*, *het-d3*, *het-e4*, *het-r*, *het-V;* E1: *het-B1*, *het-c1*, *het-d3*, *het-e1*, *het-r*, *het-V;* RV1: *het-B1*, *het-c2*, *het-d3*, *het-e4*, *het-R*, *het-V1*. Mating type is designated with—and + symbols. Note that fructifications (perithecia) form at the B1/B2 interface indicating the absence of sexual incompatibility in contrast to the C/E and RV1/rV interactions that show partial or total sterility respectively. **C.** Microscopic observation of the confrontation zone between a B1 strain expressing GFP and a B2 strain expressing RFP. The presumed anastomosis sites are marked by an arrowhead and lysed cell with an asterisk. In upper and lower right end panels, the confrontation zone between a B1 strain expressing GFP and a B1 strain expressing RFP shows normal heterokaryotic cells expressing both fluorescent markers. Each panel is 140x110 µ in size.
(PPTX)

**S2 Fig. Functional characterization of the B2 haplotype.** (A) Incompatibility phenotype of a *ΔBh2* strain. *ΔBh2* strains show a barrage reaction to B1. (B) Schematic summary of transformation efficiencies experiments of three fragments of the B2-locus region (fragments *ab*, *ac*, *db*). No transformants were obtained with fragments *db* and *ab* transformed into the B1 strain. (C) *Bp2* determines B2-incompatibility. Strains deleted for *Bp2* (*ΔBp2*) show no barrage reaction to B1 and *ΔBh1* strains transformed with *Bp2* produce a barrage reaction to B1.
(PDF)

**S3 Fig. Pairwise barrage confrontations of B1, B2, *ΔBh1*, *ΔBh2*, *ΔBp1* and *ΔBp2* strains.** Strains with the indicated genotype were grown for 5 days on corn meal agar at 26˚C.
(PDF)

**S4 Fig. Structure prediction of the BH C-terminal domain.** (A) AlphaFold models of the BH1 and BH2 lectin fold domains are given together with the structure of the *Sclerotium rolfsii* SRL lectin (pdb:2OFC). The β-strand regions are numbered from N- to C-terminus following the topology diagram given in B. In BH2 a large loop region between β-strand 4 and 5 was omitted. (B) Topology diagram of the SRL lectin (top) compared to the topology diagram derived from the BH1 lectin fold domain AlphaFold model (bottom). (C) HMMER consensus sequence of lectin fold domain of fungal BH-homologs. Above the consensus sequence, an alignment of BH1 and BH2 is given. Position of the predicted β-strands (blue arrows) and α-helix (red cylinder) are given both for the alignment and the consensus sequence.
(PDF)

**S5 Fig. Structure prediction of BP.** (A) Sequence alignment of BP1 and BP2 and a bacterial PII-like protein, AZV25447.1 from *Pseudomonas syringae*. (B) AlphaFold model of BP2. The same model is given twice, on the left to highlight polymorphic regions (the polymorphic T-loop and α1 regions highlighted in orange and red) and on the right, with a different coloring for each subunit to illustrate the predicted trimeric structure. (C) A HMMER consensus sequence for fungal BP-homologs is given together with the sequence alignment of BP1 (upper sequence) and BP2 (lower sequence). The polymorphic T-loop and α1 regions highlighted in the AlphaFold model are underlined in orange and red in the two alignments.
(PDF)

**S6 Fig. Incompatibility phenotype of chimeric *Bh* and *Bp* alleles.** Upper panels, *ΔB* strains (lacking the entire B1 haplotype) were transformed with either *Bh1* or the chimeric *Bh2*-derived alleles as given and individual transformants were assayed in barrage test in confrontation to B2 (lines of B2 tester strains are marked with an asterisk). The domain diagram of *Bh1* and the chimeric alleles are recalled. Lower panels, *ΔB* strains (lacking the entire B1 haplotype) were transformed with either *Bp2* or the chimeric *Bp1*-derived alleles as given and individual transformants were assayed in barrage test in confrontation to B1 (lines of B1 tester strains are marked with an asterisk). The domain diagram of *Bh1* and the chimeric allele are recalled. On each plate, the upper three implants are positive control for incompatibility, that is B1 strains in the upper panels and B2 strains in the lower panels.
(PDF)

**S7 Fig. Polymorphism at the *het-B* locus.** (A) Population genetic statistics in the Wageningen collection of *P. anserina* (1991–2016). Sliding-window analysis (10 kb-long with steps of 1 kb) of the first part of chromosome 3. The upper panel contains two metrics of genetic diversity: the pairwise nucleotide diversity π in red and Watterson's theta θW in blue. The lower panel depicts the Tajima's D statistic (orange). Symbols mark the position of the centromere and known *het* genes in the area. Data from Ament-Velásquez et al. (2022) [13],. (B) Architecture of the two *het-B* haplotypes is conserved across the *P. anserina* species complex. Same-sense alignments are represented with gray links, while inverted alignments are in light red. Gene exons are represented with boxes, with sense indicated by the orientation of their triangular side. The difference in gene structure between species might be due to sequencing or annotation errors, as well as real biological differences. (C) An alignment of BP1 and BP2 and their homologs in *Cercophora samala* and *Podospora setosa* is given together with an alignment of the lectin fold domain of BH1 and BH2 and their homologs from the same species. The phylogenic tree is based on the concatenated sequences of BP and the lectin fold domain of BH-homologs as given in the alignment. Sequences from *N. crassa* NCU04912 and NCU04913 are used as outgroup.
(PDF)

**S1 Table. ORF annotation in the *het-B* region.**
(XLSX)

**S2 Table. *Bh/Bp* homologs in *Sordariales*.**
(XLSX)

**S3 Table. *Bh/Bp* homologs in *Pezizomycotina*.**
(XLSX)

**S4 Table. Pfam-domain abundance and copy number variation in *Sordariales*.**
(XLSX)

**S5 Table. Genotypes and strain information of the Wageningen Collection and other members of the *P. anserina* species complex.**
(XLSX)

**S6 Table. Oligonucleotides.**
(XLSX)

**S1 Data. Source data for Figs 1C and 3B.**
(XLSX)

## Acknowledgments

CC acknowledges the assistance of Aurore Nkiliza, Pierre Helwi, Thierry Bardin, and Maeva Brito who participated to early phases of the project as Master students and Martine Sicault for assistance in the genetic mapping.

## Author Contributions

**Conceptualization:** Corinne Clavé, Sonia Dheur, Sandra Lorena Ament-Velásquez, Sven J. Saupe.

**Data curation:** Sandra Lorena Ament-Velásquez.

**Funding acquisition:** Sandra Lorena Ament-Velásquez, Sven J. Saupe.

**Investigation:** Corinne Clavé, Sonia Dheur, Sandra Lorena Ament-Velásquez, Alexandra Granger-Farbos.

**Supervision:** Sven J. Saupe.

**Visualization:** Corinne Clavé, Sonia Dheur, Sven J. Saupe.

**Writing – original draft:** Sonia Dheur, Sandra Lorena Ament-Velásquez.

**Writing – review & editing:** Corinne Clavé, Sonia Dheur, Sandra Lorena Ament-Velásquez, Alexandra Granger-Farbos, Sven J. Saupe.

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
