## [Decision Letter · Decision Letter 0]

22 Jan 2024

Dear Sven,

Thank you very much for submitting your exciting Research Article entitled 'het-B allorecognition in Podospora anserina is determined by pseudo-allelic interaction of genes encoding a HET and lectin fold domain protein and a PII-like protein' to PLOS Genetics.

The manuscript was fully evaluated at the editorial level and by three independent peer reviewers who are all experts in fungal genetics and genomics. The reviewers appreciated the attention to an important topic, and found the work well conducted and the manuscript lucid and well written.  They identified a few minor points for revision that we ask you to address in a revised manuscript.

To summarize their views:  "This is a very fine genetic analysis of het-loci in the model fungus Podospora anserina.  To my mind, it is acceptable after minor modifications."

We therefore ask you to modify the manuscript according to the review recommendations. Your revisions should address the specific points made by each reviewer.  The three reviewers all indicated that their view was minor revision/accept.

Thank you for entrusting this exciting study to PLOS Genetics!

Hope to see you at Asilomar in March,

Yours sincerely,

Joe

Joseph Heitman, MD, PhD

Academic Editor

PLOS Genetics

Eva Stukenbrock

Section Editor

PLOS Genetics

Reviewer's Responses to Questions

**Comments to the Authors:**

Reviewer #1: The research describes the identification and characterization of the het-B locus in the filamentous fungus Podospora anserina, which controls regulated cell death when two incompatible strains fuse hyphae. Nine het loci are known in P. anserina, with this being the last to be identified, here guided by genetic segregartion for positional cloning. The mechanism of action at the locus was defined through gene replacement or gene add-in experiments, to reveal a scenario of two adjacent genes required for cell death. The Bh gene encodes a protein with a classical HET domain, which the Bp gene encodes a protein similar to PII-like proteins which only to date have been characterized in bacteria as adenine sensors. For incompatibility to be detected, the combination requires the alleles of Bh from het-B1 and Bp from het-B2. Further dissection of the proteins revealed the importance of a lectin-like domain at the C-terminus of Bh important for recognition.

In the second part of the research, the authors explored how widespread the het-B genes are within P. anserina populations and more widely in other fungi. They found a skew in allelic distribution favoring the het-B1 allele. The gene pairs are found commonly in other fungi, sometimes with different families of the HET-domain adjacent to the Bp gene. These pairs are not always involved in heterokaryotic incompatibility, illustrated by Neurospora crassa containing homologs not associated with the process. Hence, the authors were able to speculate/predict an origin likely from a defense response against other dangers, that had been modified to function in allorecognition.

Last, the authors compare and contrast the allelic diversity of the nine het loci in P. anserina.

The research should be of wide appeal to those working on fungal biology and (in)compatibility systems.

Perhaps the one weakness, which is acknowledged by the reviewers, was not providing experimental evidence of physical interactions between the Bh1 and Bp2 proteins (with one control presumably to see no interaction between Bh2 and Bp1?).

Minor points:

Line 74: remove italics from ‘and’.

Line 231: capital P for ‘Pezizomycotina’.

Line 243: elsewhere these higher taxonomic groups are in italics, so probably good to check the format and be consistent throughout.

Line 249: remove italics from ‘sp.’.

Line 299: add italics to ‘V’.

Line 390: ‘molecule’ (or ‘molecular’).

Line 481: ‘has’ to ‘as’.

Line 510: remove italics from the ends of ‘NruI’ and ‘BglII’, and change ‘3,6’ to ‘3.6’.

Line 554: two times change ‘pb’ to ‘bp’.

References: the journal will not reformat these or edit them, so if you want consistent formatting and, e.g. italics on species or gene names, this could be done now.

Line 1029 and 1040: ‘HMMER’.

Reviewer #2: PGENETICS-D-23-01403

Allorecognition is defined as ability of organisms to distinguish their own tissues from those of other individuals of the same species. Allorecognition has been described in nearly all biological systems and thus is a general phenomenon that was described genetically more than 70 years for several fungal species. Fungal genetic systems have contributed substantially to the understanding of allorecognition (or heterogenetic incompatibility) and Podospora anserina is one of the genetic model organisms.

The authors are well recognized by their previous genetic analysis of incompatibility loci from P. anserina. Here, they present a thorough and detailed genetic analysis of the het-B locus, the last of nine originally described incompatibility loci in P. anserina.

Their analysis includes the genetic mapping and characterization of the het-B locus that carries two adjacent genes, which exist in divergent allelic variants in incompatible haplotypes. Using recombinant het alleles, they succeed in identifying the highly divergent C-terminal lectin fold domain of BH as determining factor of recognition specificity.

Finally, the ms compares the genomes from related Podospora species to understand het gene polymorphism in populations, which has significant evolutionary consequences on outcrossing, gene flow and speciation.

In summary, this is a very solid and well-designed genetic manuscript that definitely deserves publication.

I have only some minor comments:

For better readability I suggest, to explain the abbreviations HET- and TIR-domain in the abstract and Introduction. Not everybody of the readers is familiar with the paper of the author from 2007 (reference 42), where these domains are explained.

Further, fig. 2 could be omitted, since the fig. does not really contribute to the understanding of the genetic differences of the het loci.

Minor points:

Give references for Podospora didyma (line 238)

Reference titles are given in capital letters or in lowercase letters

Reviewer #3: I really enjoyed reading this manuscript and learned a lot from it. It identifies the genes responsible for allorecognition (non-self recognition) at the het-B locus of Podospora anserina. The great thing about the manuscript is that it is written in a very clear and engaging style that does not require readers to have much pre-existing knowledge about the topic; the ‘het’ system is introduced and explained with superb clarity. We learn that 9 het loci, governing heterokaryon incompatibility among 15 French strains of P. anserina, were originally mapped in 1967 (ref. 22). The genes responsible for the first 8 het loci were identified in a series of papers between 1990 and 2022 (line 86). Now we have the final one, het-B, so the manuscript also includes some text and figures reviewing and discussing the similarities and differences in modus operandi among the 9 het loci. As someone who does not work in this immediate field, I found the manuscript fascinating and educational and I congratulate the authors on the quality of their writing and data presentation.

As well as demonstrating (by gene deletion) that incompatibility at het-B is governed by incompatibility between alleles of two neighboring genes (Bh1 and Bp2), the manuscript also shows elegantly that the frequencies of the het-B1 and het-B2 alleles at this locus have been maintained by balancing selection for decades (Fig. 6A, Fig. S7) which has led to a trans-species polymorphism localized at the het-B locus (Fig. 6B).

I have no major comments. The Discussion of how incompatibility systems originate (particularly non-allelic systems) was interesting and I think that the authors’ arguments are logical (e.g. the inferences on Lines 412, 416, 437, 441-444).

L82, The reference to Figure 1A here seems out of place.

L160, pb should be bp.

L181. I hadn’t heard of the PII protein family before, and I found it frustrating that I didn’t know how to pronounce this name when reading the text (I wondered was it P.I.I. or P2 ?). From a literature search, I found the latter is correct and the name is often written as P(II) or with the “II” as a subscript. The name is trivial because it originates from the initial identification of the protein in 1969 as the second peak eluting from a gel filtration column, but changing the text to say P(II) or using a subscript would make it more obvious how to pronounce this name.

L231. Pezizomycotina should have uppercase P.

L286. Need a reference for the previously published sequencing data.

L365-373: Lectins bind glycans, so is it known whether the PII proteins are glycoproteins?

L418, typo – should be ”mesosynteny”.

L894, typo – should be “shortened”.

L974, typo – should be “systems”.

L974-982 and L73. Figure S1A summarizes a lot of information, but the only reference cited as the source of this information is Bernet 1967 (ref. 22). Is this correct? Incredible if true!

**Have all data underlying the figures and results presented in the manuscript been provided?**

Reviewer #1: Yes

Reviewer #2: Yes

Reviewer #3: Yes

PLOS authors have the option to publish the peer review history of their article (what does this mean?). If published, this will include your full peer review and any attached files.

Reviewer #1: No

Reviewer #2: No

Reviewer #3: No

---

## [Editor Report · Decision Letter 1]

29 Jan 2024

Dear Sven,

We are pleased to inform you that your revised manuscript entitled "het-B allorecognition in Podospora anserina is determined by pseudo-allelic interaction of genes encoding a HET and lectin fold domain protein and a PII-like protein" has been editorially accepted for publication in PLOS Genetics. Congratulations!!!

Thank you again for supporting open-access publishing; we are looking forward to publishing your work in PLOS Genetics!  I appreciate very much your entrusting this outstanding, landmark paper to PLOS Genetics!

Yours sincerely,

Joe

Joseph Heitman, MD, PhD

Academic Editor

PLOS Genetics

Eva Stukenbrock

Section Editor

PLOS Genetics

Comments from the reviewers (if applicable):

**Data Deposition**

http://datadryad.org/submit?journalID=pgenetics&manu=PGENETICS-D-23-01403R1

**Press Queries**

---

## [Editor Report · Acceptance letter]

6 Feb 2024

PGENETICS-D-23-01403R1 

het-B allorecognition in Podospora anserina is determined by pseudo-allelic interaction of genes encoding a HET and lectin fold domain protein and a PII-like protein 

Dear Dr Saupe, 

We are pleased to inform you that your manuscript entitled "het-B allorecognition in Podospora anserina is determined by pseudo-allelic interaction of genes encoding a HET and lectin fold domain protein and a PII-like protein" has been formally accepted for publication in PLOS Genetics! Your manuscript is now with our production department and you will be notified of the publication date in due course.

With kind regards,

Zsofi Zombor

PLOS Genetics

On behalf of:
